# A scalable platform for acquisition of high-fidelity human intracranial EEG with minimal clinical burden

Lisa Yamada[1,2,3], Tomiko Oskotsky[1,2], Paul Nuyujukian[1,2,3,4,5]*, for the Stanford Comprehensive Epilepsy Center[¶], Stanford Pediatric Epilepsy Center[¶]

1 Department of Bioengineering, Stanford University, Stanford, CA, United States of America, 2 Department of Neurosurgery, Stanford University, Stanford, CA, United States of America, 3 Department of Electrical Engineering, Stanford University, Stanford, CA, United States of America, 4 Wu Tsai Neurosciences Institute, Stanford University, Stanford, CA, United States of America, 5 Stanford Bio-X, Stanford University, Stanford, CA, United States of America

¶ Non-author group membership is listed in the Acknowledgments.
* 24one.plos@pn.stanford.edu

**Data Availability Statement:** All relevant data are within the paper and its Supporting information files.

## Abstract

Human neuroscience research has been significantly advanced by neuroelectrophysiological studies from people with refractory epilepsy–the only routine clinical intervention that acquires multi-day, multi-electrode human intracranial electroencephalography (iEEG). While a sampling rate below 2 kHz is sufficient for manual iEEG review by epileptologists, computational methods and research studies may benefit from higher resolution, which requires significant technical development. At adult and pediatric Stanford hospitals, research ports of commercial clinical acquisition systems were configured to collect 10 kHz iEEG of up to 256 electrodes simultaneously with the clinical data. The research digital stream was designed to be acquired post-digitization, resulting in no loss in clinical signal quality. This novel framework implements a near-invisible research platform to facilitate the secure, routine collection of high-resolution iEEG that minimizes research hardware footprint and clinical workflow interference. The addition of a pocket-sized router in the patient room enabled an encrypted tunnel to securely transmit research-quality iEEG across hospital networks to a research computer within the hospital server room, where data was coded, de-identified, and uploaded to cloud storage. Every eligible patient undergoing iEEG clinical evaluation at both hospitals since September 2017 has been recruited; participant recruitment is ongoing. Over 350+ terabytes (representing 1000+ days) of neuroelectrophysiology were recorded across 200+ participants of diverse demographics. To our knowledge, this is the first report of such a research integration within a hospital setting. It is a promising approach to promoting equitable participant enrollment and building comprehensive data repositories with consistent, high-fidelity specifications towards new discoveries in human neuroscience.

**Funding:** This work is supported by National Institute of Health (NIH) U19NS118284, Stanford Bio-X Seed Grant IIP9-104, and the Stanford Wu Tsai Neurosciences Institute. There was no additional external funding received for this study.

**Competing interests:** The authors have declared that no competing interests exist.

## Introduction

The electrical activity of the brain, electroencephalography (EEG), has become the standard assessment tool in human neuroscience [1]. It plays a valuable role in understanding the brain and neural circuits, providing opportunities to improve diagnoses and treatments of neurological disorders [2–4]. While scalp EEG is non-invasive and ubiquitous, intracranial EEG (iEEG) provides direct measurements from the brain with lower susceptibility to artifacts (e.g., sweat potentials, eye movement, and muscle activity), resulting in higher signal-to-noise ratio. There are two types of iEEG: stereotactic EEG (sEEG, Fig 1(a) and 1(b)) that can target deep cortical structures and sulci via depth electrodes and electrocorticography (ECoG, Fig 1(c) and 1(d)) that can provide a detailed 2D mapping of the brain surface via subdural grid and strip electrodes. Local field potentials of iEEG (both sEEG and ECoG) provide a unique perspective of neural activity in-between microscale (e.g., single-unit and multi-unit) and macroscale (e.g., scalp EEG and magnetoencephalography (MEG)) recordings to closely study interactions between neural populations [5, 6]. Nonetheless, due to the substantial risks of surgically implanting intracranial electrodes in the brain, iEEG is predominantly collected from people with refractory epilepsy, for whom the potential benefits of surgery outweigh the risks. For this specific patient population, iEEG is a critical component of the clinical workup to precisely localize brain tissue triggering seizures (epileptogenic zone) for epilepsy surgery planning. With successful surgery, patients can become seizure-free and experience significant improvement in their quality of life [7, 8].

While most biosignals (e.g., magnetic resonance imaging and electrocardiograms) collected for clinical purposes are sufficient to be used for research, there is growing evidence that intracranial EEG (iEEG) sampled at a higher temporal resolution than 1–2 kHz (clinical standard) may facilitate discoveries in the field of computational neuroscience [9, 10]. Higher-fidelity iEEG may not benefit current clinical practice that relies on visual inspection of EEGs by epileptologists; however, it may offer higher precision and robustness to advance quantitative

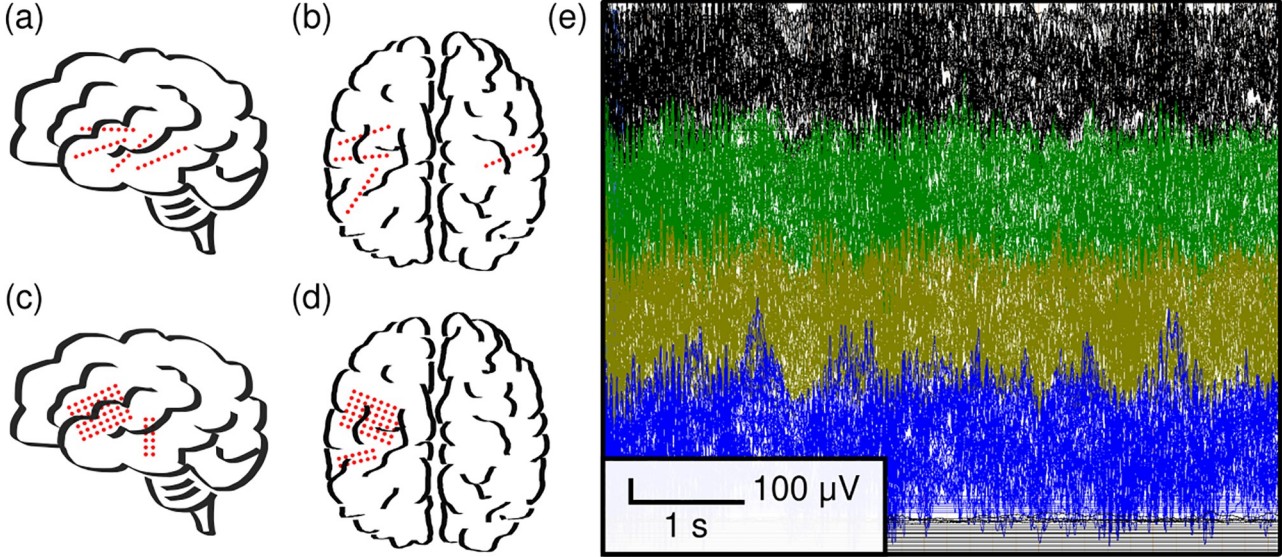

**Fig 1. Intracranial EEG (iEEG) data.** (a) Illustrations of stereo EEG with depth probes in the sagittal view. (b) Same as (a), except in the axial view. (c) Same as (a), except for electrocorticography with grid and strip electrodes. (d) Same as (c), except in the axial view. In practice, electrode quantity and spacing is solely determined by clinical need. (e) Neuroelectrophysiology sampled at 10 kHz using the Nihon Kohden JE-120A junction box. There are four intracranial electrode banks (separated by color, each with the capacity to record 64 iEEG electrodes).

tools and research studies to be translated into clinical practice [11, 12]. For instance, studies have shown that high frequency oscillations (HFOs) are important biomarkers, particularly for epilepsy, demonstrating the need to sample at much higher frequencies [13, 14]. Some studies suggest a lower bound of 2 kHz sampling rate and recommend 5 kHz for HFO analyses. It still remains a question about how fast these oscillations can reach [15, 16]; thus, this work aimed to collect iEEG with the highest sampling rate supported by our hospital-integrated platform, 10 kHz, enabling a full range of temporal activity for quantitative analyses. An example recording of 10 kHz iEEG data is shown in Fig 1(e). While a higher sampling rate requires more resources for data processing, data management tools (described in Scalable solutions for large (> 1 TB) datasets) can help facilitate the analysis and storage of these datasets.

## Methods

### Ethics statement

The clinical study associated with this effort was reviewed and approved by the Stanford Institutional Review Board and presented no more than minimal risk to participants. Participants were recruited since September 12, 2017; participant recruitment is still ongoing. Non-documented, non-witnessed, verbal, informed consent was obtained for all participants, and their parents/guardians in the case of children participants, by the research team. Study participation is voluntary and authorizes the use of research-quality iEEG data that is already simultaneously acquired with the clinical data by our proposed acquisition platform during standard of care treatment. All surgical procedures were performed independently of this study, solely based on clinical necessity; there were no interventions, changes to treatment, or impact on clinical care due to participant enrollment (e.g., electrode placement, in-patient stay duration). Due to our design considerations for our research acquisition system, apart from a brief visit for consenting during their multi-day stay (the only time researchers were physically required in the hospital), there were no changes to their neuromonitoring experience.

### Design considerations for hospital-integrated research acquisition system

Deployed in adult and pediatric Stanford hospitals, the proposed framework implements an "invisible" research platform approach with minimal research hardware in the hospital patient room, little impact on the clinical workload, and virtually no intervention from researchers during the data acquisition process. This work lays out the design strategies and technical groundwork involved in achieving a research-purpose data acquisition infrastructure that acquires high-quality iEEG, while being highly secure and minimally burdensome.

**Research EEG acquisition approach.**   Obtaining research data that is different from clinical data is not trivial in a hospital setting because it may require additional research equipment and interrupt or burden the clinical workflow. As depicted in Fig 2, there are five distinct stages to obtaining and storing clinical EEG data: the participant, clinical electrodes, analog-to-digital converter (ADC), computer, and database. At any point along this path for clinical data acquisition, research equipment can tap into the data stream to acquire EEG data for research purposes. In general, the closer the branching of the research stream is to the participant, the higher the clinical burden.

The first node in which research data can be acquired is directly from the participant's brain using research electrodes that are distinct from those used for clinical purposes [17, 18]. Of the various ways of collecting research EEG data, this method provides the largest research flexibility, as it is the only way in which researchers can decide the location to record neural activity. At the same time, it is also associated with the highest patient risk and clinical burden due to the addition of research electrodes being implanted in the brain and researcher

**Fig 2. Methods of acquiring research EEG data from the clinical stream.** The five stages for clinical iEEG acquisition are shown with different branching points for collecting research data. The branching in black (research digital stream) is utilized in this work, while the rest in gray are shown for informational purposes. From left to right there is a decrease in clinical burden across the various methods of research EEG acquisition. Abbreviations: ADC (Analog-to-digital converter).

presence in the operating room. The research electrodes have no common path with the clinical electrodes; thus, so long as sources of interference are minimized (e.g., grounding issues are addressed), the quality of the clinical stream is generally unaffected. There would, however, be no clinical copy of the iEEG data acquired for research. Additionally, research electrode placement could displace clinical electrode placement, yielding fewer clinical electrodes, but this is typically done under careful consultation with the clinical team to ensure patient care is not impacted.

Alternatively, research data can be acquired from the clinical electrodes before or after it is digitized by the clinical ADC. Both methods allow the collection of data that would not have been captured clinically (e.g., 10 kHz iEEG). However, the former requires researchers to utilize their own ADC to digitize the data, requiring expensive and sizeable research hardware in the patient room. For ease of transportability, previous studies have designed portable carts that carry all research equipment, including the ADC [10, 19–21]. Nonetheless, it is generally not optimal to have a large research footprint in restricted environments like the hospital patient room, and the need for formal acquisition sessions that require coordination and supervision by researchers and/or clinicians can interfere with clinical procedures, limiting the time with the participant and the volume of data that could be collected [8, 22]. Furthermore, adding a research ADC to the clinical data acquisition path degrades the signal quality of the clinical stream due to non-zero input impedance, parasitic draws, and inductive noise coupling. Every additional branching wire is another transmission line which draws a finite current, diverts some voltage of the clinical measurements, and introduces noise by induction of surrounding electromagnetic fields [23, 24]. All these factors come together to lower the signal-to-noise ratio of the clinical data stream, impacting recording quality. While design considerations can be made to minimally affect the clinical stream and make its impact on signal quality effectively negligible for the purposes of visual inspection, it would be preferable to avoid this concern altogether by acquiring the research data after it has been digitized by the clinical ADC (i.e., research digital stream). We believe *there is virtually no reason to tap into the clinical electrodes before digitization because using the research digital stream shares the same benefits as using a research ADC while negating its disadvantages.* Some clinical acquisition systems, such as the Nihon Kohden EEG-1200 used in this study, can be configured to output the research digital stream in an easy-to-implement and reliable manner, reducing the physical setup and presence of research equipment (removal of research ADC and accompanying hardware) and eliminating the risk of degrading the signal quality [15].

Research data that splits off of the clinical acquisition path more downstream (after being routed to the clinical computer) is limited to what was collected clinically. On the one hand,

**Table 1. Overview of various methods to acquire EEG data.**

|  | Research electrodes | Research ADC | Research digital stream | HL7 protocol | Batch copy |
|---|---|---|---|---|---|
| Clinical workflow intervention | High | Medium | Low | None | None |
| Degrade clinical signal quality | Maybe | Maybe | No | No | No |
| Physical presence in hospital | Yes | Yes | Limited | No | No |
| Restricted to clinical data | No | No | Maybe | Yes | Yes |
| Flexibility in electrode placement | Yes | No | No | No | No |
| Decreases clinical electrode count | Maybe | No | No | No | No |
| Compatible with real-time analyses | Yes | Yes | Yes | Maybe | No |
| Financial cost of research | High | High | Low | Low | Low |

Abbreviations are the same as Fig 2.

the data can be periodically transferred to research storage as it is clinically recorded via Health Level 7 (HL7) messaging protocol [25, 26]. In one study, this method enabled the collection of approximately 117 participant-years of waveform data from over 1,000 participants [27]. A complicating factor of HL7 approaches, however, is that they are not universally supported by all commercial clinical acquisition systems. On the other hand, clinical data can be batch copied for research purposes after it has already been stored in the clinical database [28]. Because this method of obtaining research data can be performed after the participant's hospital stay and without assistance from the clinical team (on the condition that appropriate permissions to access the clinical database have been granted), there is no clinical burden associated with it. Nonetheless, one disadvantage is its limited capacity to facilitate real-time analyses and feedback while the option is open for the other acquisition methods.

With careful deliberations on the pros and cons of each acquisition method (summarized in Table 1), this work deployed the acquisition method that enabled 10 kHz iEEG acquisition and optimized the balance between signal quality and clinical burden. It would not be possible to collect 10 kHz iEEG data using the HL7 protocol or copying data from the clinical database because the clinical data is typically sampled at a lower rate. Among the remaining options, obtaining the research digital stream (post-clinical ADC) was the natural choice due to having the lowest clinical burden and highest signal fidelity.

**Leveraging the existing clinical acquisition infrastructure.** In both hospitals, the clinical neuroelectrophysiology acquisition hardware used is an FDA licensed medical device, Nihon Kohden EEG-1200, which contains an electrode junction box (part number: JE-120A) that provides up to 256 neural electrodes for intracranial local field potential (LFP) recordings. This component houses the clinical and research-dedicated Ethernet ports, in which the research digital stream can be enabled by the clinical Nihon Kohden NeuroWorkbench software; once configured, the settings persist for future recording profiles. This allows the secondary research stream to sample neuroelectrophysiology at 10 kHz (i.e., the highest supported sampling rate on the JE-120A) independently and simultaneously with the clinical data stream using the same acquisition system. The ability to obtain research-quality data from the clinical system that is already present in the hospital patient room, regardless of research data collection, provides an opportunity to physically downsize the amount of research hardware involved and seamlessly integrate the research acquisition setup (shown in Fig 3) within the clinical workflow.

**Minimal physical presence of research equipment.** One of our design priorities was to reduce physical barriers for research data collection by minimizing the footprint of research hardware in the patient room (shown in Fig 4(a)) and offloading equipment to the hospital

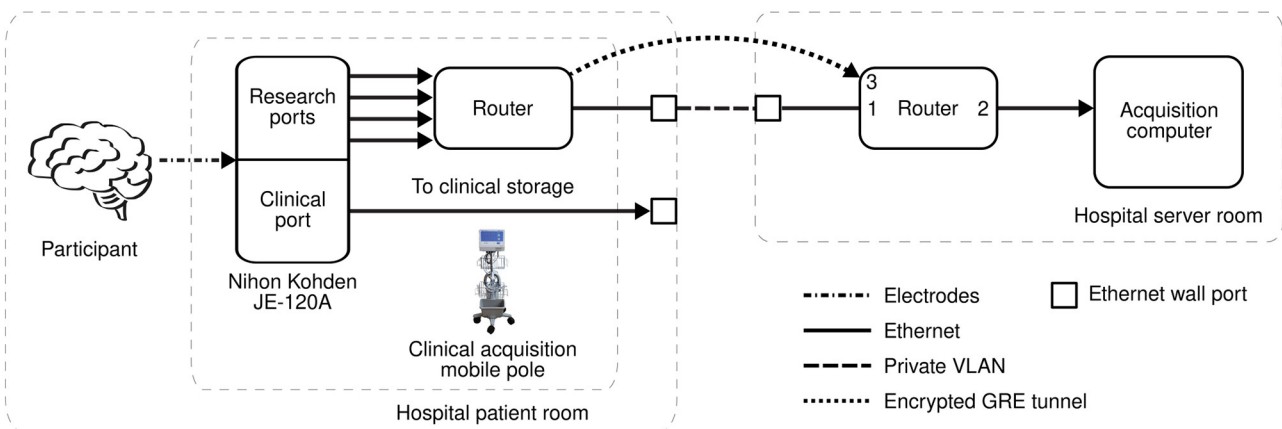

**Fig 3. Network schematic for iEEG data acquisition.** Intracranial data recorded from implanted electrodes was acquired on both clinical and research ports of the clinical acquisition system (Nihon Kohden JE-120A). The research data was sampled at 10 kHz and transmitted to the hospital server room for storage via a Generic Routing Encapsulation (GRE) tunnel between routers. Labels 1, 2, and 3 on the router in the hospital server room represent Ether1, Ether2, and the GRE tunnel, respectively, of Table 2a. Abbreviations: VLAN (virtual local area network).

server room, whenever possible. A pocket-sized router (shown in Fig 4(b)) is the only added research hardware in the patient room and it sits unobtrusively on the mobile pole, where the clinical acquisition system is installed. As shown in Fig 4(c), our research computer is racked and installed in a locked hospital server room among other clinical information technology (IT) hardware. On top of the research computer (shown in Fig 4(d)), there is an Ethernet router (hEX, MikroTik, Riga, Latvia), which is the same type as the one in the patient room. refe The routers in both hospital spaces are provisioned (see S1 Appendix) to securely transfer the acquired research-quality iEEG data from the patient room to the server room. Our unique configuration allows research hardware to be practically unseen.

**Minimal interruption to clinical workflow.** As part of the clinical acquisition mobile pole, the router (the only research hardware in the patient room) was brought into the room during clinical setup. From the electrode junction box to the router, there are semi-permanent Ethernet connections that do not need to be removed at any point, causing no additional work for the clinical team once established. Similarly, the power of the router is connected to an isolation transformer for participant safety, like the rest of the clinical equipment on the mobile pole, which also does not get disconnected in-between studies. These cables were neatly tied or selected to be about the exact length needed to keep them as tidy as possible. From the clinical team's standpoint, there is only one extra step to their setup for research purposes: plugging in an Ethernet cable between the router and Ethernet wall port. This additional Ethernet connection takes no more than thirty seconds and enables research data collection in different hospital rooms. This setup allowed researchers to start and monitor the research data collection remotely outside of the hospital and independently with no assistance from the clinical team. On occasions in which research data was unexpectedly interrupted, the clinical team was sometimes contacted to help debug the system (e.g., move the Ethernet cable to a different wall port if there was a misconfigured port).

**Health insurance portability and accountability act (HIPAA) compliance.** The original iEEG data acquired by the research ports on the clinical acquisition system contained protected health information (PHI); thus, our acquisition framework was responsible for the privacy and security of participant information to be in compliance with HIPAA [29]. Our routers deployed an encrypted Generic Routing Encapsulation (GRE) tunnel (EoIP tunnel,

... 

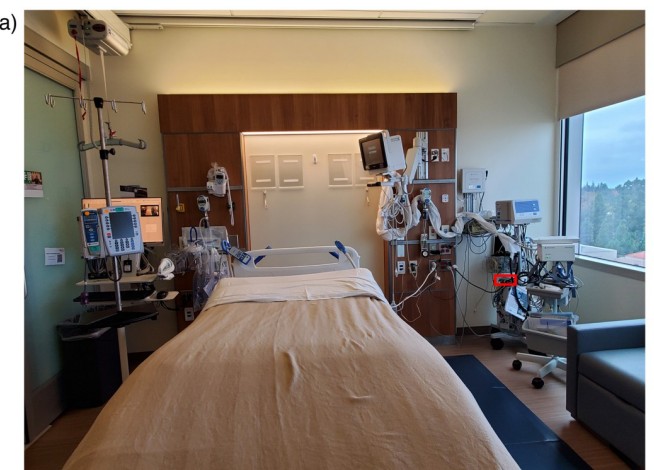
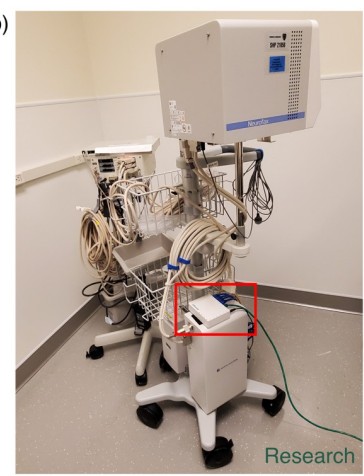
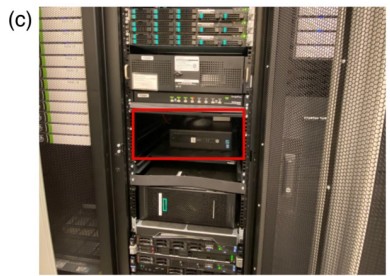
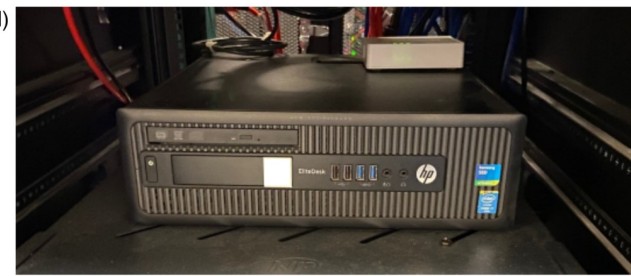

**Fig 4. Research hardware in patient room and hospital server room.** (a) A typical patient room used for iEEG neuromonitoring studies is photographed. The red box outlines where the research hardware is located. (b) The research router is placed on top of the amplifier on the clinical acquisition pole that is transported into the patient room during neuromonitoring studies. The dark green Ethernet cable is the only added connection needed from the clinical team to enable collection of the 10 kHz secondary research stream. (c) Our research computer and router is racked in the server stack in the hospital server room. (d) This is a close-up view of (c), showing only the research hardware that is part of our data infrastructure.

RouterOS v6.47, MikroTik, Riga, Latvia) based on a shared key between the patient room and server room to securely route the research data to our acquisition computer. As shown in Fig 3, this created a virtual Layer 2 bridge between specific router ports, facilitating application agnostic encryption. During data collection, there was a continuously running script in the background that immediately de-identified the transferred data when it reached the acquisition computer by programatically scrubbing PHI identifiers. The non-PHI data was then stored as part of an iEEG repository on secure cloud storage.

**Encapsulation of multiple media access control (MAC) addresses.** Modern networks, including most hospital networks, commonly have numerous safety measures to protect from malicious attacks. One common safety measure causes an Ethernet port to shut down if multiple MAC addresses are simultaneously seen from it, known as a MAC address flood guard [30]. Our research stream spans four research-dedicated Ethernet ports, each with a unique MAC address. Technically, connecting each of the four data streams to a separate wall Ethernet port would not trip the guard; however, the data would not be encrypted and it is not always the case that there would be four open Ethernet ports available in a patient room. By leveraging a router, the four data streams can be merged into one data stream and the GRE tunnel shields and encapsulates the four individual MAC addresses of the research ports from the hospital network. Thus, only the MAC address of the router is visible from the Ethernet wall port of the

patient room, allowing data to be successfully transferred to the server room. Consolidating to a single MAC address also simplifies the management of network policies for the hospital IT services to enable our platform.

**Collaboration with hospital information technology (IT) services.** As described above, the routers serve multiple purposes: reduce the physical footprint of research hardware, deploy an encrypted tunnel for PHI, and encapsulate multiple MAC addresses of the research ports. In order to employ routers within a hospital setting, coordination with hospital IT services was necessary. They assigned IP addresses for our routers and acquisition computer and configured the Identity Services Engine (ISE) policies to automatically place our equipment on a dedicated research virtual local area network (VLAN). In doing so, our research hardware communicated with each other in physically distant locations within the hospital, enabling research data collection from any hospital patient room. Furthermore, hardware support teams facilitated the physical installation of our research computer in the hospital server room.

## Scalable solutions for large (> 1 TB) datasets

The key difference between clinical and research data streams is the sampling rate. At the two hospitals, clinical data streams were collected at 1–2 kHz and stored within the hospital clinical network. While the research data contains up to ten times more data, even managing the clinical data presents challenges in terms of both computation and storage, due to the large amount of data acquired from multi-day recordings [31]. After EEG review, clinical EEG segments around seizures are often archived for long-term storage, while the duration in-between seizures–known as interictal periods–are typically deleted to conserve storage space. While of less value for epilepsy workup, EEG of interictal periods can play a critical role in seizure analyses. Thus, there is a need for long-term storage of continuous EEG with high sampling rates.

**Clinical and research data streams.** The neuroelectrophysiology data from the acquisition system is encoded as int16 values (16 bits per sample), the minimum raw bandwidth per electrode at 1 kHz is 16 kbps. The clinical stream is transferred via the clinical port with a network speed of 10 Mbps, which is more than sufficient to accommodate the routing of 1 kHz iEEG data for up to 256 electrodes and the associated network and system-level overheads. For the research stream sampled at 10 kHz, the minimum bandwidth per electrode for the raw data is 160 kbps (up to 10x the clinical stream); the added bandwidth makes it difficult to support recordings with hundreds of electrodes over a single port. Thus, the research data stream spans four research-dedicated Ethernet ports on the electrode junction box. These ports are set to autonegotiate the network speed between 10 Mbps to 1 Gbps, depending on what the hospital network can support. As shown in Fig 3, the four research streams are combined into a single stream by the router in the patient room before the 10 kHz data is transferred to a secure storage system.

**Data manipulation.** The EEG acquired from the clinical acquisition system are stored in BESA format, an open-standard binary file format–where a single file contains an hour of data. Due to the large volume of iEEG data and the limitations of a streaming file format like BESA, these files were converted to HDF5, an open-source file format that leverages data chunking and compression for fast, parallel I/O and efficiency to handle large datasets that may be too large to fit in memory or on a computer [32]. With appropriate chunk sizes, arbitrary slicing of data is processed quickly by loading only a subset of data that is actively being used into random access memory (RAM). Furthermore, HDF5 provides flexibility across platforms, is supported by many programming languages (e.g., Python, MATLAB, Java, Fortran, C, C++), increasing the ability to share and collaborate with others. In this work, two versions of HDF5 conversions were generated per iEEG recording: one version consisted of a single file

per electrode, and another version packaged all electrode information into a single file. In both cases, iEEG data of hourly BESA files were stitched together in time, and all other settings (e.g., data type, chunking and filter parameters) were kept the same. When processing individual electrodes, the single electrode files may be sufficient; however, when applying computations across multiple electrodes at once, the unified file may reduce processing time at the cost of working with a larger file.

**Data storage.** In addition to random access of data segments without loading to RAM, HDF5 supports compression filters on data chunks, allowing large datasets to be stored more compactly for lower storage costs. Furthermore, its hierarchical structure allows diverse data objects to maintain relationships by linking one another and also stores metadata directly with its relevant data. These files devoid of PHI were uploaded to cloud storage to support data access from multiple computational resources.

## Results

### Network characterization

The system network was analyzed during the data collection of an adult participant (dataset SD145, representative of a typical recording with 129 electrodes). All network measurements were made by the router in the hospital server room.

**Bandwidth.** During data acquisition, there was a steady stream of data routed from the patient room to the acquisition computer. The constant data rates are shown in Table 2a and measured from the server room router at three separate nodes labeled 1 (Ether1), 2 (Ether2), and 3 (GRE tunnel) in Fig 3. Ether1 is the physical connection between the routers in the patient room and server room and transmits encrypted data, whereas the GRE tunnel is the virtual bridge that treats the two routers as though they were directly connected and transmits the unencrypted PHI data. The overhead associated with this GRE tunnel encryption is 8.8% (see Table 2b). Ether2 is the physical connection that transmits the decrypted data from the router to the acquisition computer. Since the traffic going through the GRE tunnel and Ether2 conceptually reflect the same unencrypted stream, the data rate for the two nodes are the same at 50 Mbps, consisting of the raw data plus the network and application overhead (e.g., non-payload overhead of Open Systems Interconnection (OSI) Layers 2–7) [33]. Based on these protocol overheads, a recording at full capacity of electrodes is expected to reach slightly above 100 Mbps, which can be comfortably supported by a Gigabit Ethernet connection, which was available by both hospital networks.

**Processing power.** The routers are equipped with a dual-core quad-thread 880 MHz CPU. The idle CPU load, when research data acquisition was not enabled, was practically 0%. During data collection, Fig 5 illustrates the typical behavior of the aggregate CPU load over a 5

**Table 2. Network data rates and protocol overheads.**

| (a) | | (b) | | | |
|---|---|---|---|---|---|
| Node | Mbps | | Raw data | + Network overhead | + GRE tunnel |
| Ether1 | 54.4 | Mbps | 20.6 | 50.0 | 54.4 |
| Ether2 | 50.0 | Overhead (%) | – | 140 | 8.8 |
| GRE tunnel | 50.0 | | | | |

The data rates and protocol overheads calculated for a recording of 129 electrodes (dataset SD145). (a) The data rate measured at nodes 1 (Ether 1), 2 (Ether 2), and 3 (GRE tunnel) as indicated in Fig 3. (b) The total bandwidth observed at each router node, spanning the theoretical raw data rate, raw data + network overhead, and raw data + network overhead + GRE tunnel encryption.

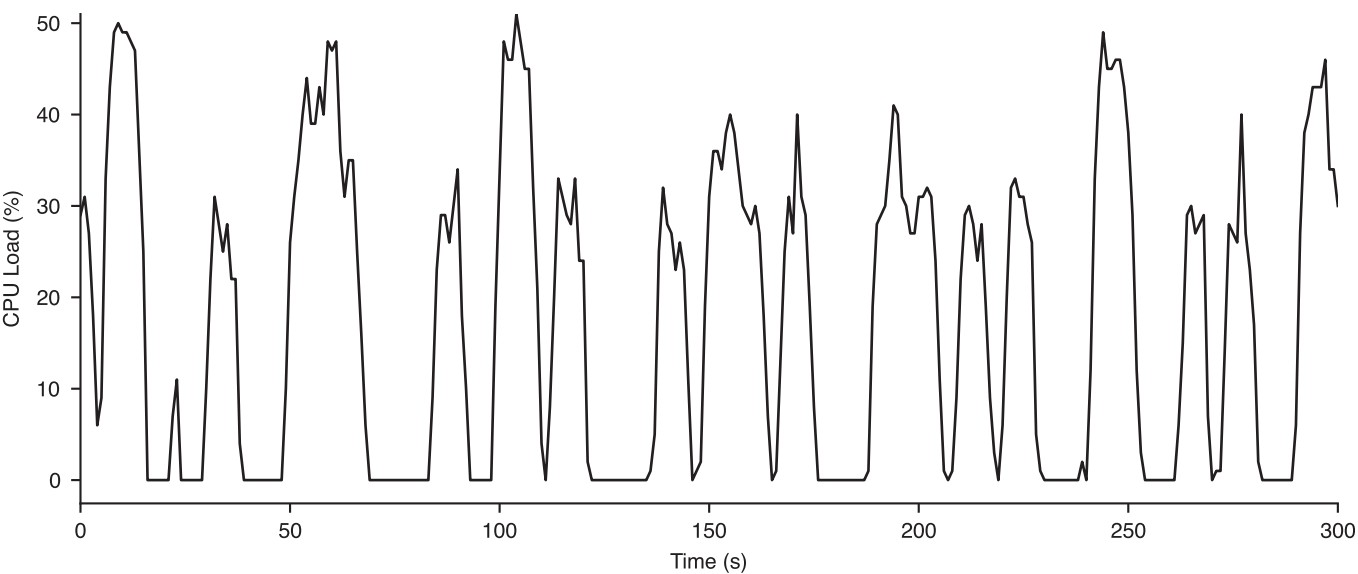

**Fig 5. Behavior of CPU load during research data collection.** The aggregate CPU load (%) of the router in the hospital server room during 5 minutes of research iEEG acquisition.

minute period. The routing and encryption of GRE tunnel traffic was offloaded to the router's dedicated Advanced Encryption Standard (AES) cipher processing hardware–not CPU–so it was not unusual that the aggregate CPU load was at 0% for more than 30% of the time. The CPU load behaved periodically; the non-zero CPU peaks are believed to be associated with periodic renegotiation of symmetric keys for the AES128 encrypted block cipher used by the GRE tunnel for security (i.e., maintain forward secrecy). Similar behavior was observed across numerous recordings.

## Storage characterization

The storage utilization was analyzed using a representative dataset of a pediatric participant PS164, in which 174 iEEG electrodes were collected over 134 hours. The storage of the raw data (excluding overhead) consisted of 1.5 TB (69 MB/electrode/hour), while the compressed datasets accounted for 0.92 TB (42 MB/electrode/hour). There were negligible differences between the two compressed formats: one file with all electrode data and individual electrode files. In this example, 38% storage savings were observed; similar storage savings were found across other datasets.

## iEEG data repository

Having deployed this platform for over six years, there have been more than 200 iEEG neuro-monitoring studies recorded across about 100 adults (49% female, 51% male) and 100 children (45% female, 55% male), ranging from 1 year to 73 years old of age. The iEEG data repository amounts to over 350 terabytes, representing 1000+ days of neuroelectrophysiology, and continues to expand over time. The HDF5 datasets are generated from the same infrastructure, resulting in consistent specifications that exceed basic technical requirements (e.g., sampling rate and number of electrodes) suggested for quantitative EEG analysis [34].

## Discussion

### Near-invisible research platform enabled by interdisciplinary expertise

Designing a hospital-integrated data infrastructure like the one presented requires meticulous deliberation among health providers and researchers to ensure participant safety, address clinical concerns, and optimize design considerations to facilitate research activities within clinical restraints [8]. By leveraging best-practices in network technology and engineering principles in collaboration with hospital IT services, a near-invisible research platform approach was realized: the patient room looked practically the same as the clinical setup while research-quality iEEG data was collected behind the scenes. The day-to-day operations of participants and clinicians were minimally impacted during iEEG evaluations; thus, the research acquisition infrastructure adapted well in the clinical environment. While this work demonstrates the acquisition of high-fidelity human iEEG, its design considerations can be utilized for collecting other types of human data in a hospital setting.

Its implementation was low cost, such that the routing of terabytes of neuroelectrophysiology per clinical study was facilitated by a single office-grade ($\sim$ \$1000) desktop and two affordable (<\$100) commercial routers. It has already been adopted in both of our institutional hospitals and could readily be scaled to multiple centers. Multiple clinical acquisition poles can be set up with the research router, such that the clinical team is not restricted to using one acquisition pole. Furthermore, this platform can theoretically extend to support the collection of multiple iEEG recordings simultaneously, although that need has thus far been rare and has not yet been empirically validated. To achieve this, new IP addresses would need to be assigned to the second set of router and clinical acquisition system. While the CPU load experienced during cipher negotiations (shown in Fig 5) may seem relatively large to support multiple streams, since they are not hard realtime operations, the CPU scheduler of the router (RouterOS is built upon a Linux kernel) should seamlessly coordinate the CPUs to generate cipher keys for each recording at different times as capacity permits.

Our research acquisition computers were securely installed in each hospital server room, which not only minimized the presence of researchers and equipment in the patient room, but also provided an ideal environment with low traffic to maintain our infrastructure. While essential research personnel can remotely connect to the acquisition computer to manage data collection, if there were issues that could only be resolved by physically interacting with it (e.g., pressing the power button), assistance from the hardware support team was requested. In terms of maintaining the acquired iEEG data, the storage savings of the compressed files can be beneficial due to the accumulating large volumes of neurophysiology collected (hundreds of terabytes).

The near-invisible design and deployment depends on our ability to leverage an easily configurable secondary research stream provided by the hospitals' commercial clinical system (Nihon Kohden, EEG-1200). Our open-source methodology may not be compatible with other iEEG clinical systems (e.g., Blackrock Microsystems and Natus) that do not accommodate research streams (if available) to have separate sampling rates from the clinical stream. While they can technically provide iEEG research data with high sampling rates up to 16 kHz (under the condition that the clinical data uses the same sampling rate), many clinical storage systems are not built to accommodate yields of terabytes per week of recording.

While it may be tempting to approach research data acquisition in the most straightforward or convenient method, sophisticated methods that minimize clinical burden and improve its scalability and maintainability may have a profound impact on the clinical and scientific community. With orchestrated efforts by clinicians, hospital technicians, and researchers, we

believe this work presents the most minimally disruptive acquisition platform for continuous 10 kHz iEEG data leveraging an existing clinical acquisition system.

## Facilitating equitable research

The minimally intrusive nature of this work, enabled by well-adopted engineering principles, significantly facilitated participant recruitment. No potential participants so far have refused to participate in the study. Since September 2017, all eligible participants–those undergoing iEEG clinical neuromonitoring studies at the hospitals during the active deployment of this infrastructure–have been recruited, regardless of age, gender, ethnicity, race, and medical condition. Thus, our participant population is diverse and is a roughly representative cross-sectional sampling of the local demographics served by the hospitals, promoting equity and justice within EEG studies. We believe our participant cohort is less subject to the typical patient demographic biases present in academic vs regional and community hospitals. Intracortical monitoring of seizure activity is a relatively rare procedure and generally not performed by regional or community hospitals. Nonetheless, access to services is still likely the largest source of bias. In the case of epilepsy, it is critical to characterize and represent the wide spectrum of EEG patterns of heterogeneous people and seizure types (e.g., semiology) [35]; however, there remains underrepresented groups in research and access to treatment (e.g., referral for epilepsy surgery evaluations, insurance coverage, and affordability) [36–38]. The data collected from this infrastructure can help support findings that better generalize well across all people, as well as identify common features within certain subgroups. Federal funding agencies (e.g., National Institutes of Health (NIH)) consistently ask about the recruiting process and push researchers to equally represent various demographic groups; as demonstrated, purposeful incorporation of engineering techniques can help facilitate equitable participant enrollment towards mitigating research gaps and addressing diagnostic and treatment options for all demographics of people.

## High-quality iEEG data repository

Our platform facilitates the acquisition of uninterrupted 10 kHz iEEG data of up to 256 neural electrodes. This is the highest sampling rate supported by the clinical system, enabling a small research footprint in the patient room. Other electrophysiological signals (e.g., electrocardiogram and blood pressure) and digital signals for tagging events/behavior were also sampled at 10 kHz and synced with the neural data. This includes sixteen channels of digital input that can be used by researchers for synchronizing neural data with external systems (e.g., conducting behavioral tasks). Furthermore, the datasets were augmented with relevant clinical annotations by epileptologists (gold standard), such as seizure times and semiology.

The high electrode count of our recordings improves spatial resolution to better capture network effects and interactions between brain regions [6, 39]. Moreover, having continuous neuroelectrophysiology throughout the entire neuromonitoring evaluation is beneficial for studying the evolution of non-stationary signals like EEG over time [40]. In epilepsy, these characteristics can help analyze seizures because they manifest from dynamic processes of neural activity that may have begun minutes, hours, or even days beforehand [41]. In fact, EEG patterns can differ significantly on the same electrode at different seizure times or nearby electrodes at the same time, emphasizing the importance of maintaining temporal and spatial relationships [42, 43].

Early EEG studies were hindered by discontinuous datasets from incomplete sets of electrodes of small number of participants [44, 45]. To overcome the EEG data shortages for research, there have been several initiatives to building a more comprehensive EEG data repository [46].

However, the distributed nature across multiple centers has resulted in a wide range of specifications (e.g., sampling rate of 250 Hz to 2.5 kHz); they could benefit from a unified, scalable framework to acquire iEEG data [47]. For reliable testing of quantitative methods for EEG studies, a standardized iEEG data repository with consistent specifications is fundamental [48, 49].

To the best of our knowledge, there is no repository of continuous iEEGs with hundreds of participants that match the data quality acquired in the clinical study enabled by the research platform presented in this work. The reason is that *collecting research-quality data different from the clinical data does not scale without technical development to the hospital infrastructure.* Without impacting signal quality, our research data acquisition approach integrates with the existing clinical infrastructure with as few changes and minimal costs as possible, such that there are minimal barriers to routinely collect from a large number of participants.

Prior work using iEEG data have demonstrated valuable findings related to human cognition, learning, and behavior–based on data that is collected during presurgical studies for people with refractory epilepsy [50–52]. However, due to the nature in which they are collected, there may be concerns for generalizability of results across all people, including healthy people and people with less severe cases of epilepsy who do not require invasive neuromonitoring. Furthermore, during clinical evaluations, participants are often put in seizure-inducing conditions (e.g., withdrawal of anti-epileptic drugs and sleep pattern alteration) and do not reflect real-life, ambulatory data.

Aside from research EEG acquisition methods that can occur alongside clinical recordings within the hospital, described in Research EEG acquisition approach, there are experimental alternatives that utilize implants to record long-term iEEG in ambulatory settings from people with refractory epilepsy [22]. One of these devices is the responsive neurostimulator (RNS), a device that monitors and stimulates brain regions in real-time to prevent the spread of seizures. However, as a fully-implanted, wireless medical device, current implementations provide limited data access for researchers. To conserve battery power, it collects short EEG clips (e.g., 90 seconds) around detected abnormalities, and has a relatively low total storage capacity ($< 60$ electrode-minutes) of EEG activity before needing to transfer the data to a secure database to free up storage. RNS devices are also typically limited by the number of electrodes from which they could record because most participants only have one or two leads (depth, subcortical strips, or combination of both) with four electrodes each [53, 54]. While other EEG recording approaches are actively in development, intracranial EEGs from seizure localization studies strike a good balance between signal quality and long-term monitoring to continue pursuing neuroscience questions [55, 56].

## Opportunities for human neuroscience

This work aims to reduce technical challenges to adapt research questions to the clinical environment for iEEG research and encourage researchers to adopt the near-invisible research platform to the greatest extent possible [8]. For instance, closed-loop neuroprosthetics and neuromodulation research may benefit from infrastructure such as this by offloading the research hardware outside the patient room [20]. While the basic framework for collecting research iEEG data is presented here, there is potential in expanding this infrastructure to behavioral neuroscience studies. In maintaining our "invisible" platform strategy, participants can choose to interface with a customized mobile device that wirelessly transfers digital signals associated with their natural or prompted behavior. These digital signals would be synced with the 10 kHz iEEG data and mark events and behavior that is recognized by the mobile device. Furthermore, this infrastructure can extend to realtime streaming of iEEG data by tapping into the digital data stream at the time of acquisition and transmitting it to a computational program or platform [57].

Furthermore, a comprehensive EEG data repository opens up opportunities for discoveries in neuroscience [44]. In a related work, our iEEG data repository collected with this platform was used to identify changes in the information content of EEG signals to demonstrate a model-free seizure detection algorithm across tens of participants of equal representation of adults and children [58]. The unified framework employed across all datasets allows batch analysis in a consistent matter and reliable algorithmic testing over a large number of participants. Separately, repository data from a participant with a unique aura was analyzed to investigate the experience of disassociation across rodents and humans, and was found to modulate homologous regions of the brain in similar ways [59]. Scalable approaches that can systematically and broadly amass higher quality iEEG data will facilitate the advancement of robust quantitative clinical metrics and human neuroscience studies.

## Supporting information

**S1 Appendix. Configuration commands for router provisioning.** These settings work against a fresh device reset of a mikrotik router and are used to provision both server and client routers. Default mikrotik IP is 192.168.88.1.
(PDF)

**S2 Appendix. CPU load data as a CSV file.** These CPU load data values are plotted in Fig 5.
(CSV)

## Acknowledgments

### Non-author group contributors

Non-author group membership followed the recommendations of the International Committee of Medical Journal Editors (ICMJE) and Council of Science Editors (CSE).
   *Stanford Comprehensive Epilepsy Center*
   Robert S. Fisher, Robert Morales, Harinder Pal Kaur, and Adam Fogarty
   *Stanford Pediatric Epilepsy Center*
   Brenda E. Porter, Jeff Mendoza, and Betty Cobb

### Special thanks

The authors acknowledge Kimberly Chin and Margaret Truong for administrative support. They also thank their study participants.

## Author Contributions

**Conceptualization:** Paul Nuyujukian.

**Data curation:** Lisa Yamada, Tomiko Oskotsky.

**Formal analysis:** Lisa Yamada.

**Funding acquisition:** Paul Nuyujukian.

**Investigation:** Lisa Yamada.

**Methodology:** Paul Nuyujukian.

**Project administration:** Paul Nuyujukian.

**Resources:** Paul Nuyujukian.

**Software:** Lisa Yamada, Tomiko Oskotsky, Paul Nuyujukian.

**Supervision:** Paul Nuyujukian.

**Validation:** Paul Nuyujukian.

**Visualization:** Lisa Yamada.

**Writing – original draft:** Lisa Yamada.

**Writing – review & editing:** Lisa Yamada, Tomiko Oskotsky, Paul Nuyujukian.

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
