## [Decision Letter · Decision Letter 0]

12 Dec 2023

PONE-D-23-31649A scalable platform for acquisition of high-fidelity human intracranial EEG with minimal clinical burdenPLOS ONE

Dear Dr. Paul Nuyujukian,

Thank you for submitting your manuscript to PLOS ONE. After careful consideration, we feel that it has merit but does not fully meet PLOS ONE’s publication criteria as it currently stands. Therefore, we invite you to submit a revised version of the manuscript that addresses the points raised during the review process.

I believe that the paper is generally well-written. However, both reviewers have mentioned the need for revisions, and I agree that modifications are necessary. Therefore, I kindly request that you make the necessary changes in accordance with the comments of the two reviewers.

We look forward to receiving your revised manuscript.

Kind regards,

Ayataka Fujimoto

Academic Editor

PLOS ONE

Journal Requirements:

"PN received support from Stanford Wu Tsai Neurosciences Institute and Stanford Bio-X Seed Grant IIP-9 104. "

3. One of the noted authors is a group or consortium Stanford Comprehensive Epilepsy Center and Stanford Pediatric Epilepsy Center. In addition to naming the author group, please list the individual authors and affiliations within this group in the acknowledgments section of your manuscript. Please also indicate clearly a lead author for this group along with a contact email address.

4. We note that Figures 3 and 4 in your submission contain copyrighted images. All PLOS content is published under the Creative Commons Attribution License (CC BY 4.0), which means that the manuscript, images, and Supporting Information files will be freely available online, and any third party is permitted to access, download, copy, distribute, and use these materials in any way, even commercially, with proper attribution. For more information, see our copyright guidelines: http://journals.plos.org/plosone/s/licenses-and-copyright.

a. You may seek permission from the original copyright holder of Figures 3 and 4 to publish the content specifically under the CC BY 4.0 license. 

Additional Editor Comments:

Two reviewers have provided their reviews. Both recommend revisions, and I find their judgments to be very fair. Please make the necessary modifications in accordance with their comments.

Reviewers' comments:

Reviewer's Responses to Questions

**Comments to the Author**

1. Is the manuscript technically sound, and do the data support the conclusions?

Reviewer #1: Yes

Reviewer #2: Yes

2. Has the statistical analysis been performed appropriately and rigorously? 

Reviewer #1: N/A

Reviewer #2: N/A

3. Have the authors made all data underlying the findings in their manuscript fully available?

Reviewer #1: No

Reviewer #2: Yes

4. Is the manuscript presented in an intelligible fashion and written in standard English?

Reviewer #1: No

Reviewer #2: Yes

5. Review Comments to the Author

Reviewer #1: Dr. Yamada et al. reported a methodology and system for recording intracranial EEG data of drug-resistant epilepsy patients’ evaluations, with 10kHz iEEG up to 256 electrodes simultaneously, along with clinical data. As mentioned in the manuscript, higher-fidelity iEEG has merits in accurately localizing the epileptogenic zone and using the data for understanding physiological human neuroscience. In this context, their suggested methodology offers minimal invasiveness to clinical courses and patients, as well as higher temporal resolution data. The authors described the background in the introduction and the flow of intracranial EEG data with clinical burden, as shown in Figure 1. This makes it easier for readers who are not familiar with the detailed methodology to understand the principles of their suggested methodology and system. Their description in the introduction and method not only justifies their research and emphasizes its clinical impact but also makes the manuscript easy to follow.

However, my concern with publishing this article as a “Research Article” is the lack of statistical comparison with a control. Generally speaking, methodology papers are required to be compared with the conventional method to demonstrate their validity as a new method or system.

Therefore, I suggest the authors consider either of the following options:

(1) Submitting as “Submissions describing methods, software, databases, or other tools.” Please find the details of the article types in PLOS ONE: https://journals.plos.org/plosone/s/journal-information#loc-scope.

(2) Attaching data that shows the validity of the suggested methodology or system with a comparison to the conventional method.

Additionally, I suggest describing the weaknesses of this method. For example, 10kHz recording holds ten times more information than 1kHz, which means it requires ten times more data space and longer analysis time.

Reviewer #2: Yamada et al. reported their technical approach for building comprehensive data repositories of more than 200 participants of diverse demographics. They mentioned that their promising approach is promoting equitable enrollment and building comprehensive data repositories with consistent, high-fidelity specifications towards new discoveries in human neuroscience.

Comments (invitation: November 28, 2023, and submission: November 29, 2023)

1) We want to congratulate the authors’ efforts. The manuscript is well written and easy to follow. By establishing a research platform that minimizes clinical burden, it becomes feasible to record data from a substantial number of participants while maintaining consistency. None of the following comments are criticisms.

2) Introduction: Would it be possible to provide the reasons behind the acquisition of 10 kHz iEEG data? The authors mentioned in this article that 5 kHz is recommended (line 34). It would be nicer to include the authors' perspective on this matter in the Discussion section (High-quality iEEG data repository subsection).

3) Methods and Figure1: Would it be possible to specify which stage in Figure 1 corresponds to the method employed in this study? Figure 1, as described by the authors, delineates five distinct stages to obtaining and storing clinical EEG data (line 53 and 54), with each stage elaborated on in the methods section. It would be nicer to provide a visual representation within Figure 1 to highlight the method used by the authors.

4) Methods and Figure2: To explain “the integration of our research data infrastructure in a hospital setting” (line122 and 123), Figure 2 is being used; however, I wondered if Figure 2 might not be necessary for this paper. If it is to be used, adding some explanation on how the integration leads to the generation of Figure 2 would make it more understandable for readers.

5) Results: Would it be possible to provide the difference in the number of research papers produced at the authors' facility before and after the actual use of this platform. It would be helpful for readers to understand the difference.

6) Figure 1, 3 and Table 1: It would be nicer to include the explanations of the abbreviations in the legend.

6. PLOS authors have the option to publish the peer review history of their article (what does this mean?). If published, this will include your full peer review and any attached files.

Reviewer #1: No

Reviewer #2: **Yes: **Kazuki Sakakura

---

## [Decision Letter · Decision Letter 1]

9 Apr 2024

A scalable platform for acquisition of high-fidelity human intracranial EEG with minimal clinical burden

PONE-D-23-31649R1

Dear Dr. Nuyujukian,

We’re pleased to inform you that your manuscript has been judged scientifically suitable for publication and will be formally accepted for publication once it meets all outstanding technical requirements.

Kind regards,

Ayataka Fujimoto

Academic Editor

PLOS ONE

Additional Editor Comments (optional):

This is a very well-written paper. With this revision, I have endorsed this paper.

Reviewers' comments:

Reviewer's Responses to Questions

**Comments to the Author**

1. If the authors have adequately addressed your comments raised in a previous round of review and you feel that this manuscript is now acceptable for publication, you may indicate that here to bypass the “Comments to the Author” section, enter your conflict of interest statement in the “Confidential to Editor” section, and submit your "Accept" recommendation.

Reviewer #1: All comments have been addressed

Reviewer #2: All comments have been addressed

2. Is the manuscript technically sound, and do the data support the conclusions?

Reviewer #1: Yes

Reviewer #2: Yes

3. Has the statistical analysis been performed appropriately and rigorously? 

Reviewer #1: N/A

Reviewer #2: No

4. Have the authors made all data underlying the findings in their manuscript fully available?

Reviewer #1: Yes

Reviewer #2: Yes

5. Is the manuscript presented in an intelligible fashion and written in standard English?

Reviewer #1: Yes

Reviewer #2: Yes

6. Review Comments to the Author

Reviewer #1: I appreciate the authors taking my suggestions and incorporating them into the manuscript.

I would like to congratulate their effort in the revision.

Reviewer #2: The authors have replied sufficiently to all my comments. This is a very nice manuscript. Kazuki Sakakura

7. PLOS authors have the option to publish the peer review history of their article (what does this mean?). If published, this will include your full peer review and any attached files.

Reviewer #1: No

Reviewer #2: **Yes: **Kazuki Sakakura

---

## [Editor Report · Acceptance letter]

4 Jun 2024

PONE-D-23-31649R1 

PLOS ONE

Dear Dr. Nuyujukian, 

I'm pleased to inform you that your manuscript has been deemed suitable for publication in PLOS ONE. Congratulations! Your manuscript is now being handed over to our production team.

Kind regards, 

on behalf of

Dr. Ayataka Fujimoto 

Academic Editor

PLOS ONE